# INFERENCE SCALING FOR LONG-CONTEXT RETRIEVAL AUGMENTED GENERATION

**Zhenrui Yue**[*†1]**, Honglei Zhuang**[*2]**, Aijun Bai**[2]**, Kai Hui**[2]**, Rolf Jagerman**[2]**, Hansi Zeng**[†3]
**Zhen Qin**[2]**, Dong Wang**[1]**, Xuanhui Wang**[2]**, Michael Bendersky**[2]
[1]University of Illinois Urbana-Champaign, [2]Google DeepMind, [3]UMass Amherst
zhenrui3@illinois.edu, hlz@google.com

## ABSTRACT

The scaling of inference computation has unlocked the potential of long-context large language models (LLMs) across diverse settings. For knowledge-intensive tasks, the increased compute is often allocated to incorporate more external knowledge. However, without effectively utilizing such knowledge, solely expanding context does not always enhance performance. In this work, we investigate inference scaling for retrieval augmented generation (RAG), exploring the combination of multiple strategies beyond simply increasing the quantity of knowledge, including in-context learning and iterative prompting. These strategies provide additional flexibility to scale test-time computation (e.g., by increasing retrieved documents or generation steps), thereby enhancing LLMs' ability to effectively acquire and utilize contextual information. We address two key questions: (1) How does RAG performance benefit from the *scaling of inference computation* when optimally configured? (2) Can we predict the optimal *test-time compute allocation* for a given budget by modeling the relationship between RAG performance and inference parameters? Our observations reveal that increasing inference computation leads to nearly linear gains in RAG performance when optimally allocated, a relationship we describe as the *inference scaling laws for RAG*. Building on this, we further develop the *computation allocation model* to estimate RAG performance across different inference configurations. The model predicts optimal inference parameters under various computation constraints, which align closely with the experimental results. By applying these optimal configurations, we demonstrate that scaling inference compute on long-context LLMs achieves up to 58.9% gains on benchmark datasets compared to standard RAG.

## 1 INTRODUCTION

Long-context large language models (LLMs) are designed to handle extended input sequences, enabling them to process and understand longer context (e.g., Gemini 1.5 Pro with up to 2M tokens) (Achiam et al., 2023; Team et al., 2023; Reid et al., 2024). Combined with increased inference computation, long-context LLMs demonstrate improved performance across various downstream tasks (Agarwal et al., 2024; Snell et al., 2024). For example, many-shot in-context learning (ICL) can match the performance of supervised fine-tuning by providing extensive in-context examples (Bertsch et al., 2024). Particularly for knowledge-intensive tasks that leverage retrieval augmented generation (RAG), increasing the quantity or size of retrieved documents up to a certain threshold consistently enhances the performance (Ram et al., 2023; Xu et al., 2024; Jiang et al., 2024).

Previous studies on inference scaling for RAG focus on expanding the retrieved knowledge by increasing the number or lengths of retrieved documents (Xu et al., 2024; Jiang et al., 2024; Shao et al., 2024). However, only emphasizing on the knowledge quantity without providing further guidance presents certain limitations. On one hand, current long-context LLMs still have limited ability to effectively locate relevant information in ultra-long sequences upon challenging tasks (Li et al., 2024; Kuratov et al., 2024). For instance, the optimal performance of long-context LLMs

---

*Equal contribution
†Work done while at Google DeepMind

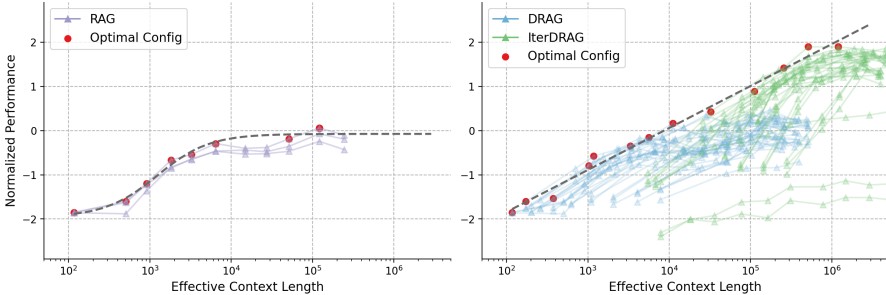

Figure 1: Normalized performance vs. effective context lengths on MuSiQue. Each line represents a fixed configuration, scaled by adjusting the number of documents. Red dots and dash lines represent the optimal configurations and their fitting results. Standard RAG plateaus early at $10^4$ tokens, in contrast, DRAG and IterDRAG show near-linear improvement as the effective context length grows.

is often achieved without fully utilizing the maximum length (Agarwal et al., 2024). On the other hand, numerous studies show that retrieving over soft thresholds (e.g., top-10 documents) leads to a performance plateau and may even cause declines (Ram et al., 2023; Lee et al., 2024a; Kuratov et al., 2024). Such performance drops may be traced back to the increased noise within context, which causes distraction and adversely affects generation (Yoran et al., 2024; Zhang et al., 2024; Leng et al., 2024). As a result, inference scaling of long-context RAG remains challenging for existing methods.

In this work, we leverage a broader range of strategies to comprehensively explore how RAG benefits from the scaling of inference computation. A straightforward strategy is *demonstration-based RAG* (DRAG), where multiple RAG examples are provided as demonstrations to utilize the long-context capabilities of LLMs (Brown et al., 2020). DRAG allows models to learn (in-context) how to locate relevant information and apply it to response generation[1]. Nevertheless, the quality of one-step retrieval varies across tasks and often fails to provide sufficient information. Inspired by iterative methods (Trivedi et al., 2023; Yoran et al., 2024), we develop *iterative demonstration-based RAG* (IterDRAG). IterDRAG learns to decompose input queries into simpler sub-queries and answer them using interleaved retrieval. By iteratively retrieving and generating upon sub-queries, LLMs construct reasoning chains that bridge the compositionality gap for multi-hop queries. Together, these strategies provide additional flexibility in scaling inference computation for RAG, allowing long-context LLMs to more effectively address complex knowledge-intensive queries.

Building on these strategies, we investigate multiple ways to scale up inference computation. Here, we measure computation by considering the total number of input tokens across all iterations, referred to as the *effective context length*. In DRAG, scaling the effective context length can be done by increasing two inference parameters: the number of retrieved documents and in-context examples. In IterDRAG, test-time compute can be further extended by introducing additional generation steps. Since different combinations of inference parameters result in varied allocations of computational resources, our goal is to establish the relationship between RAG *performance*, different *scales* and *allocations* of inference computation. Through extensive experiments on benchmark QA datasets, we demonstrate an almost linear relationship between RAG performance and the scale of effective context length by combining both RAG strategies, as shown in Figure 1 (right). Moreover, our RAG strategies exhibit improved performance than merely scaling the number of documents, achieving state-of-the-art performance with the compact Gemini 1.5 Flash (See evaluation in Figure 2).

Drawing from our observations, we examine the relationship between RAG performance and inference computation, which we quantify as the *inference scaling laws for RAG*. These observed inference scaling laws reveal that RAG performance consistently improves with the expansion of the effective context length under optimal configurations. Consequently, we take a deeper dive into modeling RAG performance with respect to various inference computation *allocations*. Our goal is to predict the optimal set of inference parameters that maximize the performance across different RAG tasks. To achieve this, we quantitatively model the relationship between RAG performance and varying inference configurations with the *computation allocation model for RAG*. Using the

---

[1]Different from in-context RAG that prepends documents / QA examples (Press et al., 2023; Ram et al., 2023), we leverage multiple examples comprising of documents, questions and answers to demonstrate the task.

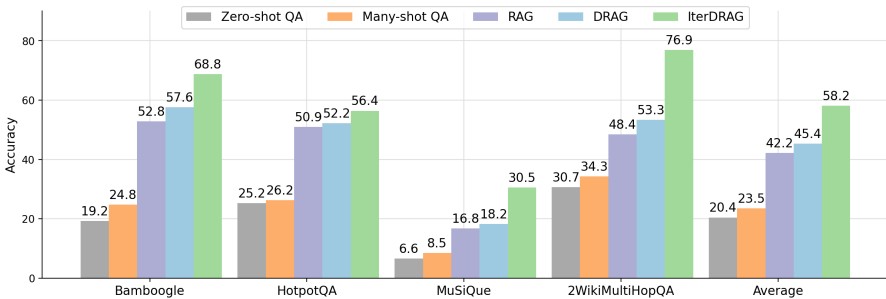

Figure 2: Evaluation accuracy of Gemini 1.5 Flash using different methods: zero-shot QA, many-shot QA, RAG (with an optimal number of documents), DRAG and IterDRAG on benchmark QA datasets. By scaling up inference compute (up to 5M tokens), DRAG consistently outperforms baselines, while IterDRAG improves upon DRAG through interleaving retrieval and iterative generation.

estimated computation allocation model, the optimal configurations can be empirically determined and generalize well for various scenarios, thereby maximizing the utilization of the computation budget. We summarize our contributions as follows:

- We systematically investigate inference scaling for long-context RAG, for which we introduce two scaling strategies, DRAG and IterDRAG, to effectively scale inference compute.

- We comprehensively evaluate DRAG and IterDRAG, where they not only achieve state-of-the-art performance, but also exhibit superior scaling properties compared to solely increasing the quantity of documents.

- Through extensive experiments on benchmark QA datasets, we demonstrate that when test-time compute is optimally allocated, long-context RAG performance can scale almost linearly with the increasing order of magnitude of the computation budget.

- We quantitatively model the relationship between RAG performance and different inference parameters, deriving the computation allocation model. This model aligns closely with our experimental results and generalize well across scenarios, providing practical guidance for optimal computation allocation in long-context RAG.

## 2 RELATED WORK

### 2.1 LONG-CONTEXT LLMS

Long-context large language models (LLMs) are designed to utilize extensive context and thereby improve their generative capabilities. Early works in extending context lengths involve sparse / low-rank kernels to reduce memory requirements (Kitaev et al., 2019; Beltagy et al., 2020; Zaheer et al., 2020; Choromanski et al., 2020). In addition, recurrent and state space models (SSMs) are proposed as efficient substitutes for transformer-based models (Gu et al., 2021; Gu & Dao, 2023; Peng et al., 2023a; Beck et al., 2024). For causal LLMs, extrapolation and interpolation methods have proven effective in expanding context window lengths (Press et al., 2021; Chen et al., 2023; Sun et al., 2023; Peng et al., 2023b). Recent advancements in efficient attention methods (Dao et al., 2022; Jacobs et al., 2023; Liu et al., 2023) further enable LLMs to train and infer upon input sequences comprising millions of tokens (Achiam et al., 2023; Team et al., 2023; Reid et al., 2024).

### 2.2 IN-CONTEXT LEARNING

In-context learning (ICL) offers a computationally efficient approach to enhance model performance at inference time by conditioning on a few demonstrations of the task (Brown et al., 2020). To further improve ICL performance, existing works focuses on pretraining strategies that optimize the language models to learn in-context (Min et al., 2022; Wei et al., 2023; Gu et al., 2023). In addition, selective usage of few-shot examples are shown to be helpful for enhancing downstream task performance (Liu et al., 2022; Rubin et al., 2022; Wang et al., 2024). Notably, reformatting or finding optimal ordering of in-context examples also improves ICL performance effectiveness (Lu et al., 2022; Wu et al., 2023;

Liu et al., 2024a). With the emergence of long-context LLMs (Achiam et al., 2023; Team et al., 2023; Reid et al., 2024), scaling the number of examples becomes possible in ICL (Li et al., 2023; Bertsch et al., 2024; Agarwal et al., 2024). For instance, Agarwal et al. (2024) show that many-shot ICL can mitigate pretraining biases within LLMs and thus improves ICL performance across various tasks.

## 2.3 RETRIEVAL AUGMENTED GENERATION

Retrieval augmented generation (RAG) improves language model performance by incorporating relevant knowledge from external sources (Lewis et al., 2020; Guu et al., 2020; Karpukhin et al., 2020). In contrast to naïve RAG, optimizing the retrieval stage can effectively enhance context relevance and improve generation performance (Ma et al., 2023; Trivedi et al., 2023; Jiang et al., 2023; Shi et al., 2024; Sarthi et al., 2024; Lin et al., 2024; Mo et al., 2024). An example is REPLUG, in which Shi et al. (2024) leverage LLM as supervision to learn a dense retriever model. In addition, encoding documents can increase knowledge retrieval and improve generation capabilities (Khandelwal et al., 2019; Izacard & Grave, 2021; Borgeaud et al., 2022; Izacard et al., 2023). For instance, Izacard & Grave (2021) leverages fusion-in-decoder architecture to encode multiple question-passage pairs while maintaining the model efficiency. Alternatively, selectively utilizing knowledge from the documents improves the robustness of LLMs against irrelevant context (Yu et al., 2023; Yoran et al., 2024; Yan et al., 2024; Yue et al., 2024; Zhang et al., 2024). For example, RAFT proposes to train language models with negative documents to improve generation quality and relevance (Zhang et al., 2024). Concurrent to our work, long-document retrieval and datastore scaling are proposed to optimize RAG performance (Jiang et al., 2024; Shao et al., 2024). Despite such progress, inference scaling remains under-explored for long-context RAG methods. As such, we investigate how variations in inference computation impact RAG performance, with the goal of optimizing test-time compute allocation.

## 3 INFERENCE SCALING STRATEGIES FOR RAG

### 3.1 PRELIMINARIES

We measure inference computation with *effective context length*, defined as the total number of input tokens across all iterations before the LLM outputs the final answer. For most methods that only call the LLM once, the effective context length is equivalent to the number of input tokens in the prompt and is limited by the context window limit of the LLM. For methods that iteratively call the LLM, the effective context length can be extended indefinitely depending on the strategy. We exclude output tokens and retrieval costs from our analysis, as LLMs typically generate significantly fewer tokens (fewer than 10) in knowledge-intensive tasks. Additionally, retrieval is generally much less computationally expensive than LLM inference, especially with scalable matching methods (Sun et al., 2024). Our objective is to understand how RAG performance changes as we scale up inference computation. In demonstration-based RAG (DRAG), we achieve such scaling by incorporating both extensive documents and in-context examples. For further scaling, we increase generation steps through iterative demonstration-based RAG (IterDRAG). We introduce both strategies below.

### 3.2 DEMONSTRATION-BASED RAG

Demonstration-based RAG (DRAG) leverages in-context learning to exploit the capabilities of long-context LLMs by directly generating answers from an extended input context. DRAG builds upon naïve RAG and integrates both documents and in-context examples into the input prompt. This expanded context allows the model to generate answers to the input query within a single inference request (See Figure 3 left). For both in-context examples and the test-time query, we employ a retrieval model to select the top-$k$ retrieved documents from a large corpus (e.g., Wikipedia). We reverse the order of the retrieved documents, placing higher-ranked documents closer to the query (Liu et al., 2024b). As we use instruction-tuned LLMs, we design a similar prompt template following Agarwal et al. (2024) and align the formatting with prefixes for retrieved documents, input and output (See Appendix I). Unlike previous works (Press et al., 2023; Trivedi et al., 2023), DRAG incorporates extensive retrieved documents within the demonstrations, enabling long-context LLMs to learn to extract relevant information and answer questions using a rich input context.

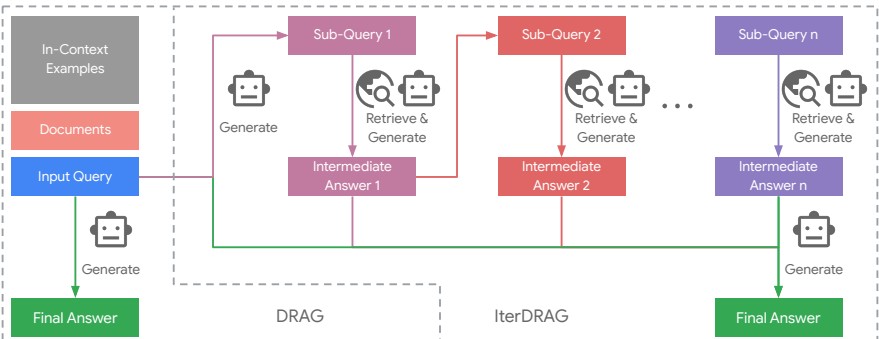

Figure 3: DRAG vs. IterDRAG. IterDRAG breaks down the input query into sub-queries and answer them to improve the accuracy of the final answer. In test-time, IterDRAG scales the computation through multiple inference steps to decompose complex queries and retrieve documents.

### 3.3 ITERATIVE DEMONSTRATION-BASED RAG

Despite access to external knowledge, complex multi-hop queries remain challenging due to the compositionality gap. To tackle this issue, we introduce iterative demonstration-based RAG (Iter-DRAG), which handles complex queries by decomposing the query into simpler sub-queries. For each sub-query, retrieval is performed to gather additional contextual information, which is then used to generate intermediate answers. After all sub-queries are resolved, the retrieved context, sub-queries, and their answers are combined to synthesize the final answer (See Figure 3 right).

While multiple existing datasets provide training data with queries and corresponding answers, sub-queries and intermediate answers are often absent. To generate in-context examples with sub-queries and intermediate answers, we prompt LLMs with constrained decoding to follow the Self-Ask format (Press et al., 2023; Koo et al., 2024). In each iteration, LLMs generate either a sub-query, an intermediate answer, or the final answer. If a sub-query is generated, additional documents are retrieved and interleaved into the prompt before producing the intermediate answer. IterDRAG continues until the final answer is generated or the number of maximum iterations is reached, at which point LLM is forced to generate the final answer. We retain examples with intermediate steps and correct final answers to construct in-context demonstrations. Each example should include the retrieved documents, sub-query and answer pairs, as well as the final answer.

During inference, in-context examples are prepended to the initial documents retrieved for the input query. Similarly, each inference request yields a sub-query, an intermediate answer, or the final answer. Upon sub-queries, additional documents are retrieved and merged with the initial ones to generate intermediate answers. In our implementation, we allow up to five iterations of query decomposition before generating the final answer. This iterative process effectively scales test-time computation, with the input tokens from all iterations summed to calculate the effective context length. IterDRAG facilitates a more granular approach by learning to: (1) decompose query into simple and manageable sub-queries; and (2) retrieve and locate relevant information to answer (sub-)queries. As such, the iterative retrieval and generation strategy helps narrowing the compositionality gap and improves knowledge extraction, thereby enhancing overall RAG performance.

## 4 RAG PERFORMANCE AND INFERENCE COMPUTATION SCALE

### 4.1 FIXED BUDGET OPTIMAL PERFORMANCE

For a given budget on inference computation, i.e., a maximum effective context length $L_{\max}$, there are multiple ways to optimize the use of computation resources through inference parameters. For example, in DRAG, we can adjust both the number of retrieved documents and in-context examples, while in the IterDRAG strategy, we additionally introduce the number of iterations for retrieval and generation. Henceforth, we use $\theta$ to denote all these inference parameters.

Table 1: Optimal performance of different methods with varying maximum effective context lengths $L_{max}$ (i.e., the total number of input tokens *across all iterations*). ZS QA and MS QA refers to zero-shot QA and many-shot QA respectively. Partial results are omitted for methods that do not further scale with increasing $L_{max}$. For clarity, we mark the best results for each $L_{max}$ in bold.

| $L_{max}$ | Method | Bamboogle | | | HotpotQA | | | MuSiQue | | | 2WikiMultiHopQA | | |
|---|---|---|---|---|---|---|---|---|---|---|---|---|---|
| | | EM | F1 | Acc | EM | F1 | Acc | EM | F1 | Acc | EM | F1 | Acc |
| 16k | ZS QA | 16.8 | 25.9 | 19.2 | 22.7 | 32.0 | 25.2 | 5.0 | 13.2 | 6.6 | 28.3 | 33.5 | 30.7 |
| | MS QA | 24.0 | 30.7 | 24.8 | 24.6 | 34.0 | 26.2 | 7.4 | 16.4 | 8.5 | 33.2 | 37.5 | 34.3 |
| | RAG | 44.0 | 54.5 | 45.6 | 44.2 | 57.9 | 49.2 | 12.3 | 21.5 | 15.3 | 42.3 | 49.3 | 46.5 |
| | DRAG | 44.0 | 55.2 | 45.6 | **45.5** | **58.5** | 50.2 | **14.5** | **24.6** | **16.9** | 45.2 | 53.5 | 50.5 |
| | IterDRAG | **46.4** | **56.2** | **51.2** | 36.0 | 47.4 | 44.4 | 8.1 | 17.5 | 12.2 | 33.2 | 38.8 | 43.8 |
| 32k | RAG | 48.8 | 56.2 | 49.6 | 44.2 | 58.2 | 49.3 | 12.3 | 21.5 | 15.3 | 42.9 | 50.6 | 48.0 |
| | DRAG | **48.8** | **59.2** | 50.4 | **46.9** | **60.3** | **52.0** | **15.4** | **26.0** | 17.3 | 45.9 | 53.7 | 51.4 |
| | IterDRAG | 46.4 | 56.2 | 52.0 | 38.3 | 49.8 | 44.4 | 12.5 | 23.1 | **19.7** | 44.3 | 54.6 | 56.8 |
| 128k | RAG | 51.2 | 60.3 | 52.8 | 45.7 | 59.6 | 50.9 | 14.0 | 23.7 | 16.8 | 43.1 | 50.7 | 48.4 |
| | DRAG | 52.8 | 62.3 | 54.4 | **47.4** | **61.3** | 52.2 | 15.4 | 26.0 | 17.9 | 47.5 | 55.3 | 53.1 |
| | IterDRAG | **63.2** | **74.8** | **68.8** | 44.8 | 59.4 | **52.8** | **17.3** | **28.0** | **24.5** | **62.3** | **73.8** | **74.6** |
| 1M | DRAG | 56.0 | 62.9 | 57.6 | 47.4 | 61.3 | 52.2 | 15.9 | 26.0 | 18.2 | 48.2 | 55.7 | 53.3 |
| | IterDRAG | **65.6** | **75.6** | **68.8** | **48.7** | **63.3** | **55.3** | **22.2** | **34.3** | **30.5** | **65.7** | **75.2** | **76.4** |
| 5M | IterDRAG | **65.6** | **75.6** | **68.8** | 51.7 | 64.4 | 56.4 | 22.5 | 35.0 | 30.5 | 67.0 | 75.2 | 76.9 |

For each input query and its ground-truth answer $(x_i, y_i) \in \mathcal{X}$, we can apply the RAG inference strategy $f$ parameterized by $\theta$. We denote the effective input context length to the LLM as $l(x_i; \theta)$ and the obtained prediction as $\hat{y}_i = f(x_i; \theta)$. A metric $P(y_i, \hat{y}_i)$ can then be calculated based on $y_i$ and $\hat{y}_i$. To understand the relationship between RAG performance and inference computation, we sample a few different inference computation budgets. For each budget $L_{max}$, we find the optimal average metric $P^*(L_{max})$ achievable within this budget by enumerating different $\theta \in \Theta$:

$$P^*(L_{max}) := \max_{\theta \in \Theta} \left\{ \frac{1}{|\mathcal{X}|} \sum_i P\big(y_i, f(x_i; \theta)\big) \Big| \forall i, l(x_i; \theta) \leq L_{max} \right\}. \tag{1}$$

Our goal is to establish the relationship between the inference computation budget $L_{max}$ and the best possible performance within this budget $P^*(L_{max})$, using any possible strategies and parameter configurations to allocate the inference computation resources. For simplicity, we also refer to $P^*(L_{max})$ as the *optimal performance*. We investigate the following factors within the inference parameter set $\theta$: (1) the number of documents $k$, which are retrieved from a large corpus (e.g., Wikipedia) based on the input query; (2) the number of in-context examples $m$, where each of the examples consists of $k$ documents, an input query and its label; and (3) the number of generation iterations $n$. In DRAG, an answer can be directly generated upon input context, so $n = 1$. In contrast, IterDRAG involves multiple steps of interleaved retrieval and generation, expanding both the effective context length and inference compute without needing longer context windows.

We evaluate the performance of Gemini 1.5 Flash with context length window up to 1M tokens on knowledge-intensive question answering, including multi-hop datasets Bamboogle, HotpotQA, MuSiQue and 2WikiMultiHopQA (Press et al., 2023; Yang et al., 2018; Trivedi et al., 2022; Ho et al., 2020). Additional results are provided in Appendix C and Appendix D. To manage the computational costs of extensive experiments, we follow Wu et al. (2024); Gutiérrez et al. (2024) and sample 1.2k examples from each dataset for evaluation. The evaluation metrics include exact match (EM), F1 score (F1) and accuracy (Acc), in which the accuracy metric assesses whether the ground truth is located within the prediction. We sample the inference computation budget $L_{max}$ as 16k, 32k, 128k, 1M and 5M tokens. For the parameter space $\Theta$ of DRAG, we consider the number of documents $k \in \{0, 1, 2, 5, 10, 20, 50, 100, 200, 500, 1000\}$, and the number in-context examples $m$ ranging from $0, 2^0, 2^1, ...,$ to $2^8$. For IterDRAG, we further experiment with number of iterations $n$ up to 5. We compare to the following baselines: (1) zero-shot QA (QA), where the model does not leverage any retrieved documents or demonstrations; (2) many-shot QA (MS QA), where the model only uses varying number of demonstrations $m$ without any retrieved document; and (3) retrieval augmented generation (RAG), where the model only uses $k$ retrieved documents without demonstrations. We report the optimal performance of each method with different maximum effective context length budgets by examining their performance with different inference parameter configurations.

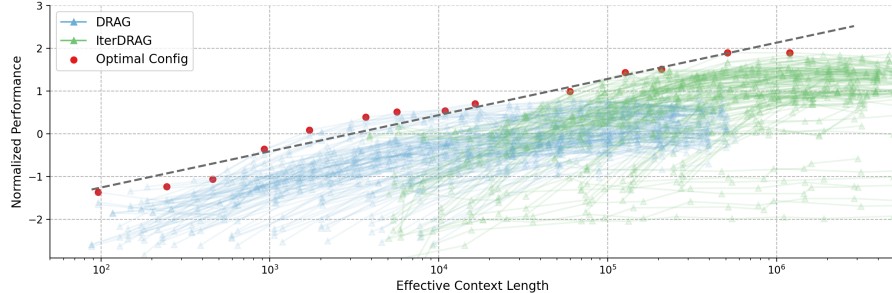

Figure 4: Normalized performance vs. effective context lengths across datasets. Each line represents a fixed configuration, scaled by varying the number of documents. Red dots indicate the optimal configurations, with the dashed line showing the fitting results. The observed optimal performance can be approximated by a linear relationship with the effective context lengths.

## 4.2 OVERALL PERFORMANCE

We report the optimal performance $P^*(L_{\max})$ for different inference strategies in Table 1, where we identify the optimal inference parameters for each computation budget $L_{\max}$. Some variants are omitted for certain $L_{\max}$ because they do not scale to the corresponding context length. For example, the prompt for zero-shot QA cannot be increased, while the number of in-context examples for many-shot QA is capped at $2^8$, so neither scales to $L_{\max} = 32$k. Similarly, RAG does not scale to $L_{\max}$ larger than 128k, and DRAG is limited by the LLM's context window limit of 1M.

Unlike QA and RAG baselines, the performance of DRAG and IterDRAG consistently increase as we expand the maximum effective context length. More specifically, we observe: (1) *DRAG and IterDRAG scale better than baselines.* Baselines like many-shot QA peak at 16k tokens, while RAG improves until 128k, after which performance plateaus. In comparison, DRAG and IterDRAG can find optimal configurations to more effectively utilize test-time compute, exhibiting superior performance and scaling properties. Performance of DRAG consistently improves until 1M tokens, while IterDRAG further enhances RAG performance with 5M tokens of computation budget by iteratively calling LLMs. (2) *DRAG excels with shorter maximum lengths, while IterDRAG scales more effectively with longer effective context length.* At 16k and 32k, DRAG typically delivers the best performance, while at 128k and beyond, IterDRAG achieves superior results overall, highlighting the effectiveness of iterative retrieval and generation. These results suggest that increasing $L_{\max}$ is beneficial for RAG, with DRAG and IterDRAG strategies each excelling at different scales.

## 4.3 INFERENCE SCALING LAWS FOR RAG

To analyze RAG performance with different effective context lengths, we plot the performance of all configurations across datasets in Figure 4. Similar to Figure 1, we visualize DRAG and IterDRAG and highlight the optimal performance $P^*(L_{\max})$ for different selections of $L_{\max}$. The fitting results are shown as grey dashed lines. We provide additional dataset-specific results in Appendix F.

The optimal performance exhibits consistent gains as the effective context length expands, demonstrating a strong linear correlation, which we term the *inference scaling laws for RAG*. Combined with dataset-specific results, our key observations are: (1) *The optimal performance scales nearly linearly with the order of magnitude of the inference compute.* Such linear relationship suggests that RAG performance can be improved by increasing computation, allowing for more accurate predictions of performance given available compute resources. (2) *For $L_{max}$ above $10^5$, IterDRAG continues to scale effectively with interleaving retrieval and iterative generation.* This aligns with our results in Table 1, where IterDRAG better utilizes computation budgets for effective context lengths exceeding 128k. (3) *Gains on optimal performance gradually diminish beyond an effective context length of 1M.* Despite dataset variations, the performance follows similar trends up to 1M tokens. Beyond that, improvements from 1M to 5M are less substantial or plateau, potentially due to limitations in long-context modeling. In summary, while gains are smaller beyond 1M tokens, optimal RAG performance scales almost linearly with increasing inference compute through DRAG and IterDRAG.

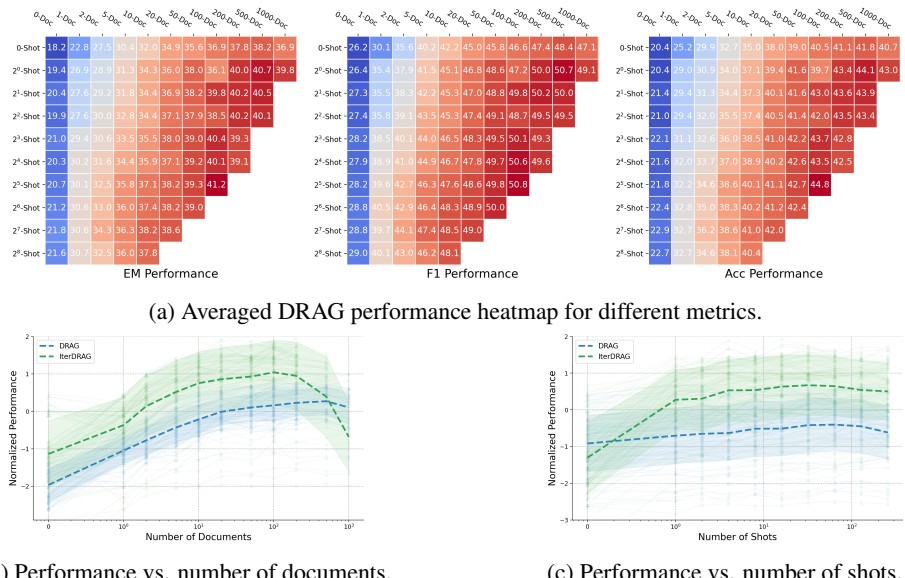

(a) Averaged DRAG performance heatmap for different metrics.

(b) Performance vs. number of documents.

(c) Performance vs. number of shots.

Figure 5: RAG performance changes with varying number of documents and in-context examples. 5a reports the averaged metric values across datasets, whereas in 5b and 5c, each line represents the normalized performance of a consistent configuration with progressively increasing documents / shots.

## 4.4 PARAMETER-SPECIFIC SCALING

To gain further insights into the dynamics of DRAG and IterDRAG, we grid search over different combinations of $\theta$ and evaluate the performance. The results are presented in Figure 5, where we visualize DRAG performance using heatmaps (See IterDRAG heatmap in Appendix D). Additionally, we provide further results with varying numbers of documents ($k$) and shots ($m$). In summary, scaling retrieval, demonstrations and more generation steps leads to performance gains in most cases, yet such gains vary by effective context length and method. In particular, we note: (1) *Documents and in-context examples are not equally helpful.* For a fixed configuration, increasing the number of retrieved documents $k$ usually leads to more substantial performance gains, as evidenced by the differing slopes in Figure 5. (2) *Increasing shots $m$ is more helpful for IterDRAG.* For example, increase $m$ from 0 to 1 (rather than $k$) is more helpful for IterDRAG, possibly due to demonstrations that leads to improved in-context query decomposition and knowledge extraction. (3) *Scaling saturates differently for DRAG and IterDRAG.* An example can be found in the increase of $m$ from 0 to 1, which results in significant improvements for IterDRAG but shows little impact on DRAG. Beyond the soft thresholds, further increases in $k$ or $m$ yield marginal gains or even results in performance declines. (4) *For a given $L_{max}$, the optimal $\theta$ depends on the method, metric and dataset.* As illustrated in Figure 5a and Figure 8, the optimal combinations are sensitive to the metrics and located differently, posing challenges for performance modeling w.r.t. $\theta$. In conclusion, increasing documents, demonstrations and iterations can enhance RAG performance, but each contributes differently to the overall results. As such, identifying the optimal combination of hyperparameters remains challenging.

## 5 INFERENCE COMPUTATION ALLOCATION FOR LONG-CONTEXT RAG

After examining the overall performance of different RAG strategies and the varying impacts of different inference parameters, we now quantify the relationship between performance and the hyperparameter set $\theta$. We hypothesize that for long-context RAG, we can model such test-time scaling properties and term it *computation allocation model for RAG*. This model, in turn, can be used to guide the selection of $\theta$ based on the maximum effective context length $L_{\max}$.

## 5.1 FORMULATION AND ESTIMATION

With a slight abuse of notation, we redefine the average performance metric $P$ (e.g., accuracy) on dataset $\mathcal{X}$ as a function of $\theta$. We consider the number of documents $k$, demonstrations $m$ and

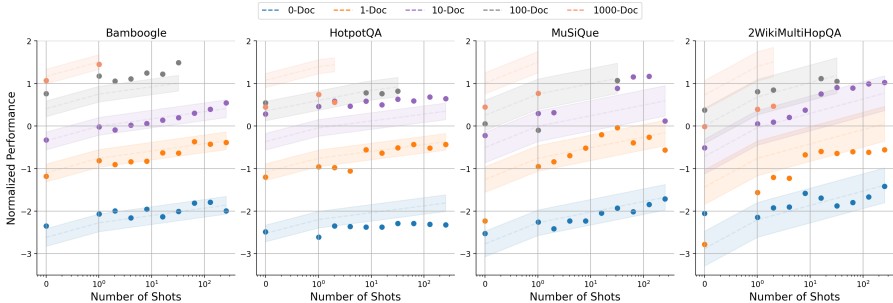

Figure 6: The estimated performance using the proposed observational scaling laws vs. actual metric values in DRAG. The subplots represent different datasets, where each line corresponds to a fixed number of documents, we scale the context length by increasing the number of shots.

maximum iterations $n$ within $\theta$, namely $\theta := (k, m, n)^T$. To account for the variance across methods and tasks, we introduce $i := (i_{doc}, i_{shot}, 0)^T$. $i_{doc}$ and $i_{shot}$ measure the informativeness of documents and in-context examples respectively. While technically we can also define an $i_{iter}$ to measure the informativeness of additional generation steps, applying $i_{iter}$ does not yield improved accuracy, so we leave it as 0 in our experiments. We formulate the computation allocation model as[2]:

$$\sigma^{-1}(P(\theta)) \approx (a + b \odot i)^T \log(\theta) + c, \tag{2}$$

where $\odot$ refers to element-wise product. $a, b \in \mathbb{R}^3$ and $c \in \mathbb{R}$ are parameters to be estimated, and $i$ can be computed base on the specific task. There are different ways to define $i$; we choose a definition to compute $i$ based on the performance difference between selected base configurations. In particular, for each strategy on each dataset, $i_{doc}$ is defined as the performance gain by only adding one document compared to zero-shot QA. Similarly, $i_{shot}$ is defined as the performance gain by adding only one in-context example compared to zero-shot QA. To account for the sub-linearity in extremely long contexts (above 1M), we apply an inverse sigmoidal mapping $\sigma^{-1}$ to scale the values of the metric $P$. Further implementation details are reported in Appendix I.

In Equation (2), estimations on $a$, $b$ and $c$ are specific to a certain model, reflecting how LLMs improve with varying number of documents and shots (i.e., in-context learning / zero-shot capabilities). In contrast, $i$ models the performance variations within the selected task (i.e., how external knowledge / demonstrations help responding to the query). Therefore, the computation allocation model can be estimated once and applied to various downstream tasks without requiring additional calibration. To estimate the parameters, varying combinations of $\theta$ are evaluated to perform ordinary least squares on $a$, $b$ and $c$. We report the parameters for Gemini 1.5 Flash in Appendix G.

## 5.2 VALIDATING THE COMPUTATION ALLOCATION MODEL FOR RAG

We evaluate the computation allocation model for RAG by comparing the predicted metrics to the actual values, with normalized results for DRAG visualized in Figure 6. Here, each subplot represents a different dataset, and each line corresponds to a document setting ($k$), we scale the context length by adjusting in-context examples ($m$). As illustrated, the performance improves with the increase of $k$ and $m$ across datasets, displaying highly consistent trends between the predicted and actual metric values, despite some variations. Notably, each dataset exhibits different levels of consistency: Bamboogle exhibits the highest consistency, while HotpotQA generates more variable results. Our findings demonstrate how external knowledge and in-context learning can effectively enhance RAG performance with long-context capabilities, suggesting the effectiveness of the computation allocation model for RAG and how they may be used to predict benchmark results.

Table 2: Ablation study results of the computation allocation model for RAG.

| | Exclude $b$ | | Quadratic $\theta$ | | Linear $\sigma$ | | Sigmoidal $\sigma$ | |
| --- | --- | --- | --- | --- | --- | --- | --- | --- |
| | $R^2$ | MSE | $R^2$ | MSE | $R^2$ | MSE | $R^2$ | MSE |
| Values | 0.866 | 0.116 | 0.867 | 0.117 | 0.876 | 0.109 | **0.903** | **0.085** |

---

[2]In our implementation, we shift the values within $\theta$ by a small $\epsilon$ to prevent numerical issues with $\log(0)$.

**Ablation Study.** To verify the effectiveness of the computation allocation model, we perform ablation studies and evaluate the fitting performance of different variants. In particular, we assess: (1) estimation without $b$ and $i$ (i.e., Exclude $b$); (2) a quadratic form of input $\log(\theta)$ (Quadratic $\theta$); (3) linear scaling of $P$ (Linear $\sigma$); and (4) sigmoid scaling of $P$ (Sigmoidal $\sigma$). The $R^2$ and MSE values for these variants are reported in Table 2, in which (4) represents the complete design of our computation allocation model. The results indicate that incorporating the additional $b$ with $i$ enhances the relevance and reduces error across all tasks. Moreover, applying inverse sigmoid to $P$ significantly improves the estimation in comparison to quadratic $\theta$ or linear scaling.

Table 3: Domain generalization results of the computation allocation model for RAG.

|  | Bamboogle | | | HotpotQA | | | MuSiQue | | | 2WikiMultiHopQA | | |
|---|---|---|---|---|---|---|---|---|---|---|---|---|
|  | EM | F1 | Acc | EM | F1 | Acc | EM | F1 | Acc | EM | F1 | Acc |
| Baseline | 49.6 | 58.8 | 51.2 | 46.3 | 60.2 | 51.4 | 14.9 | 24.7 | 16.9 | 46.5 | 53.7 | 51.6 |
| Predict | **64.0** | **75.6** | **68.0** | **47.8** | **63.3** | **55.3** | **19.3** | **32.5** | **29.3** | **60.8** | **72.4** | **74.9** |
| Oracle | 65.6 | 75.6 | 68.8 | 48.7 | 63.3 | 55.3 | 22.2 | 34.3 | 30.5 | 65.7 | 75.2 | 76.4 |

**Domain Generalization.** We also examine the generalization of the computation allocation model for RAG for unseen domains. In other words, the parameters of Equation (2) are tested on the target domain but learnt from the remaining domains. For inference, only $i$ is derived from the target domain. We report the results for 1M effective context length in Table 3, where we compare to an 8-shot baseline configuration (scaled by increasing retrieved documents) and the optimum results (Oracle). In summary, the results show that computation allocation model significantly outperforms baseline and closely aligns with the oracle results (96.6% of the optimal performance). Notably, Bamboogle and HotpotQA exhibit highly similar target results, with the performance metrics varying by less than 2.5% from the oracle. These results suggest the potential of applying the computation allocation model for RAG to a wider range of knowledge-intensive tasks.

Table 4: Length extrapolation results of the computation allocation model for RAG.

|  | 16k → 32k | | | 32k → 128k | | | 128k → 1M | | | 1M → 5M | | |
|---|---|---|---|---|---|---|---|---|---|---|---|---|
|  | EM | F1 | Acc | EM | F1 | Acc | EM | F1 | Acc | EM | F1 | Acc |
| Baseline | 37.4 | 47.6 | 40.4 | 39.0 | 49.5 | 42.2 | 39.3 | 49.3 | 42.8 | 44.5 | 55.4 | 49.8 |
| Predict | **37.4** | **48.2** | **41.0** | **41.2** | **52.0** | **45.4** | **48.0** | **60.9** | **56.9** | **47.9** | **59.8** | **55.2** |
| Oracle | 39.2 | 49.8 | 42.7 | 46.9 | 59.0 | 55.1 | 50.5 | 62.1 | 57.7 | 51.7 | 62.6 | 58.1 |

**Length Extrapolation.** In addition to predictability on unseen domains, we explore the extrapolation of context length based on the computation allocation model. Here, we estimate the parameters of Equation (2) using experiments with shorter context lengths and assess their predictive accuracy on longer ones. We assess different extrapolation settings and present the predicted metric values in Table 4. Our observations are: (1) The predictions are accurate and consistently outperform the 8-shot baseline. For instance, the average difference between the predicted and oracle results from 128k to 1M tokens is just 2.8%. (2) Extrapolating from 32k to 128k is challenging. This is because DRAG performs best around 32k, while IterDRAG typically excels at a long context of 128k, as evidenced in Figure 4. Consequently, it creates a discrepancy between training and predicting performance distribution. (3) 5M context length is less predictable, with the average performance difference between predicted and oracle metrics observed at a substantial 5.6%. Overall, length extrapolation with computation allocation model is accurate and more effective for target lengths below 1M.

# 6 CONCLUSION

In this paper, we explore inference scaling in long-context RAG. By systematically studying the performance with different inference configurations, we demonstrate that RAG performance improves almost linearly with the increasing order of magnitude of the test-time compute under optimal inference parameters. Based on our observations, we derive inference scaling laws for RAG and the corresponding computation allocation model, designed to predict RAG performance on varying hyperparameters. Through extensive experiments, we show that optimal configurations can be accurately estimated and align closely with the experimental results. These insights provide a strong foundation for future research in optimizing inference strategies for long-context RAG.

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

## A DISCUSSION

**Retrieval.** One critical factor in improving performance of RAG lies in the quality of the retrieved documents. To study how retrieval impacts final accuracy, we analyze retrieval performance and report the results across different document sizes in Appendix B. In all datasets, recall scores demonstrate improvements as the number of documents increases, approaching near-perfect scores with large document sets (e.g., ~1k). Despite consistent gains in recall, the results show diminishing returns on discounted ranking metrics like NDCG, indicating increasing distraction within the context. This trend is also evident in in Figure 5b, where RAG performance peaks between 100 and 500 documents. Our observations suggest the necessity of refining retrieval (e.g., through re-ranking) to further optimize the document relevance, particularly in cases of complex, multi-hop queries. However, how the inference scaling behavior discovered in this paper would change in the presence of such a refining component remains unknown. Alternatively, iterative retrieval, as seen in IterDRAG, improves recall performance by using simpler, straightforward sub-queries to collect additional context for each intermediate answer. In summary, retrieving more documents improves recall but does not necessarily lead to better generation quality if the documents are not effectively ranked or filtered. This highlights the need for retrieval methods that dynamically adjust to minimize irrelevant content.

**Error Analysis.** Despite overall improvements, our error analysis in Appendix H reveals that certain errors persist, particularly in cases of compositional reasoning tasks where multiple hops of reasoning are required. The common errors fall into four categories: (1) inaccurate or outdated retrieval; (2) incorrect or lack of reasoning; (3) hallucination or unfaithful reasoning; and (4) evaluation issues or refusal to answer. The first category highlights the need for enhancing retrieval methods and maintaining a reliable & up-to-date knowledge base, specially for complex questions that rely on multiple supporting facts. In addition, incorrect or missing reasoning steps often result in errors or partially correct answers. In our experiments, we observe that both (1) and (2) are substantially improved with IterDRAG, suggesting the importance of interleaving retrieval and iterative generation for multi-hop queries. Moreover, developing faithful LLMs and strategies to mitigate hallucination could further enhance RAG performance. Finally, we note that existing metrics fail in certain cases (e.g., abbreviations), underscoring the need for more robust and reliable evaluation methods.

**Long-Context Modeling.** We also discuss the impact of long-context modeling w.r.t. RAG performance. In summary, we find that retrieving more documents is generally beneficial for RAG performance, as demonstrated in Section 4. Nevertheless, naïvely extending the context length in each generation step does not always lead to better results. Specifically, DRAG performance peaks at around $10^5$ tokens, while IterDRAG achieves optimal performance at around $10^6$ tokens by leveraging multiple rounds of generation. For instance, as seen in the performance plateau in Figure 1 and Figure 11, LLMs struggle to effectively utilize very long contexts ($\geq 10^5$ tokens) in each iteration, potentially due to inherent limitations of long-context modeling. Our observations suggest that: (1) the model's ability to identify relevant information from extensive context remains to be improved, especially when presented with large quantity of "similar" documents; (2) the long-context modeling should be further refined to enhance in-context learning capabilities, where multiple lengthy demonstrations are provided.

**Trade-Off Between Inference Compute and RAG Performance.** In our experiments, we observe consistent benefits of inference scaling using DRAG and IterDRAG, potentially changing the optimal trade-off between inference compute and RAG performance. Existing methods often exhibit diminishing returns when scaling inference compute beyond certain thresholds where RAG performance plateaus. As a result, the optimal trade-off between inference compute and RAG performance is unlikely to be found beyond these thresholds, as further investment in scaling inference compute becomes inefficient. In contrast, our findings demonstrate that long-context RAG performance can improve almost linearly with increased test-time compute when optimally allocated. Therefore, the optimal trade-off in our setting largely depends on the inference budget, with higher budgets consistently yielding steady gains. Combined with the computation allocation model for RAG, this approach enables the derivation of a (nearly) optimal solution for long-context RAG given computation constraints.

# B    RETRIEVAL QUALITY

We assess the retrieval quality of DRAG and IterDRAG using the Gecko-1B model (Lee et al., 2024b) and evaluate their impact on final RAG performance. Specifically, we retrieve varying numbers of documents per input query and measure the retrieval quality using three metrics: Recall, NDCG, and MRR, with document counts ranging from 1 to 2k. The retrieval results of DRAG are shown in Figure 7. In addition, we evaluate the quality of iterative retrieval, where a maximum of five interleaving retrieval steps are performed. Here, we retrieve 50 documents at each step and use a 2-shot setting, with the results in comparison to DRAG in Table 5.

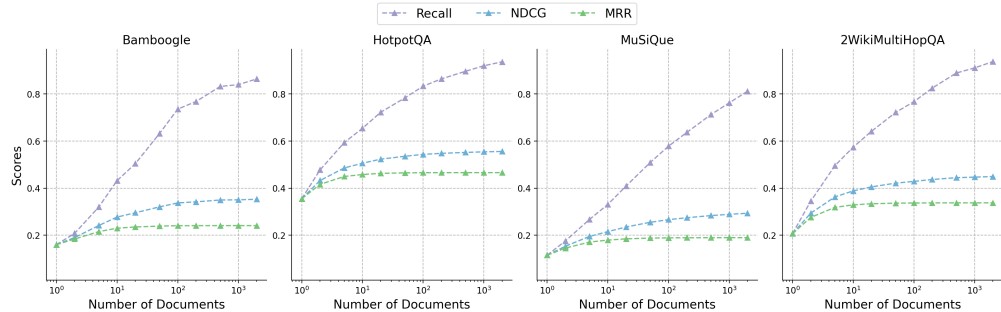

Figure 7: Retrieval performance of DRAG on different datasets.

In Figure 7, recall demonstrates consistent improvements as the number of documents increases, approaching near-perfect scores when large document sets (e.g., 1k) are retrieved. However, both NDCG and MRR metrics plateau early at around 100 documents, with diminishing gains as the document count further rises. This divergence suggests that while more documents lead to better recall, the relevance and ranking quality (captured by NDCG and MRR) do not improve proportionally, and even introduce extensive noise. Therefore, higher recall doesn't necessarily translate into better final answer quality when the retrieved documents aren't effectively ranked or filtered.

Table 5: Retrieval performance of DRAG and IterDRAG ($k = 50$ documents, $m = 2$ shots).

|  | **Bamboogle** | | | **HotpotQA** | | | **MuSiQue** | | | **2WikiMultiHopQA** | | |
|---|---|---|---|---|---|---|---|---|---|---|---|---|
|  | Recall | NDCG | MRR | Recall | NDCG | MRR | Recall | NDCG | MRR | Recall | NDCG | MRR |
| DRAG | 0.632 | 0.321 | 0.239 | 0.783 | 0.535 | 0.465 | 0.509 | 0.255 | 0.188 | 0.722 | 0.421 | 0.336 |
| IterDRAG | **0.736** | **0.420** | **0.346** | **0.855** | **0.549** | **0.478** | **0.670** | **0.365** | **0.291** | **0.935** | **0.605** | **0.528** |

Unlike the one-step retrieval in DRAG, iterative retrieval based on query decomposition often yields simpler sub-queries, facilitating more effective retrieval. In addition, merging the retrieved documents from different steps typically results in higher overall retrieval performance, as evidenced in Table 5. With IterDRAG, the performance gains are consistent and reach the average of 30.5%. Specifically, we observe higher gains for complex multi-hop queries (e.g., 2WikiMultiHopQA), where metric improvements can be as high as 57.1%. Moreover, the gains on ranking-discounted metrics (30.7% in NDCG and 39.9% MRR) show greater improvements compared to recall (21.7%). In summary, these findings highlight the superiority of iterative retrieval with query decomposition over one-step methods, which effectively contribute to the overall performance of IterDRAG.

# C    CHAIN-OF-THOUGHT VS. ITERDRAG.

Table 6: Chain-of-thought (CoT) vs. IterDRAG results ($k = 5$ documents, $m = 4$ shots).

|  | **HotpotQA** | | | **MuSiQue** | | | **2WikiMultiHopQA** | | |
|---|---|---|---|---|---|---|---|---|---|
|  | EM | F1 | Acc | EM | F1 | Acc | EM | F1 | Acc |
| CoT | 40.2 | 51.3 | 45.6 | 8.9 | 16.1 | 10.8 | 33.0 | 37.9 | 36.7 |
| IterDRAG | **44.8** | **59.4** | **52.8** | **17.9** | **30.1** | **25.9** | **57.5** | **69.9** | **72.3** |

To evaluate different iterative strategies, we compare the commonly used chain-of-thought (CoT) with IterDRAG (Wei et al., 2022). In particular, we generate the CoT examples following Trivedi et al. (2023) and adopt the 4-shot setting with 5 documents. The results on three larger datasets (HotpotQA, MuSiQue and 2WikiMultiHopQA), as reported in Table 6, highlight the performance differences between these strategies, in which IterDRAG consistently outperforms CoT with significant improvements. Such difference can be traced back to three key factors: (1) the retrieval quality of CoT is limited without interleaving retrieval as in IterDRAG; (2) Gemini 1.5 Flash is relatively small and may not perform well in free-form reasoning in comparison to larger LLMs; and (3) the generated CoT examples are less informative than handcrafted ones and underperform compared to constrained decoding with Self-Ask (Press et al., 2023; Koo et al., 2024). Consequently, IterDRAG demonstrates its effectiveness as a scalable method for knowledge-intensive tasks.

## D  ADDITIONAL RAG RESULTS

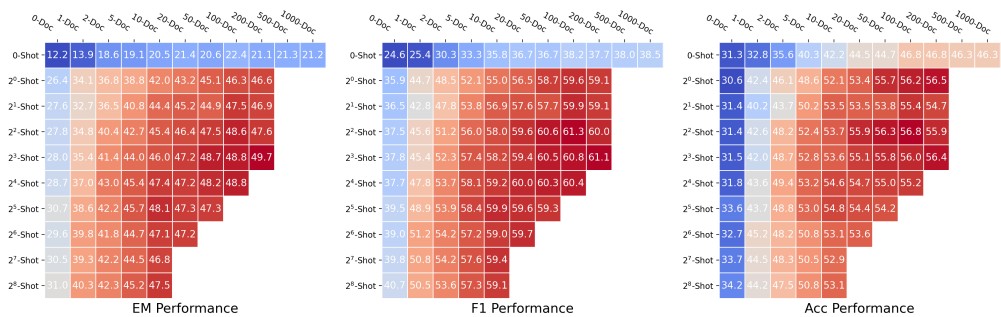

Figure 8: IterDRAG performance heatmap for different metrics averaged across datasets.

We report the IterDRAG results averaged across datasets in Figure 8, shown as heatmaps where the x-axis represents the number of documents and the y-axis represents the number of shots. Performance is color-coded, with blue indicating lower values and red indicating higher values. The best-performing combinations are located toward the bottom right of each heatmap, which corresponds to longer context lengths. In comparison to DRAG, as reported in Figure 5a, the optimal number of in-context examples is higher at 32, which highlights the importance of in-context demonstrations in enabling better query decomposition and interleaved retrieval. Combined with multiple generation steps, IterDRAG further improves RAG performance over DRAG.

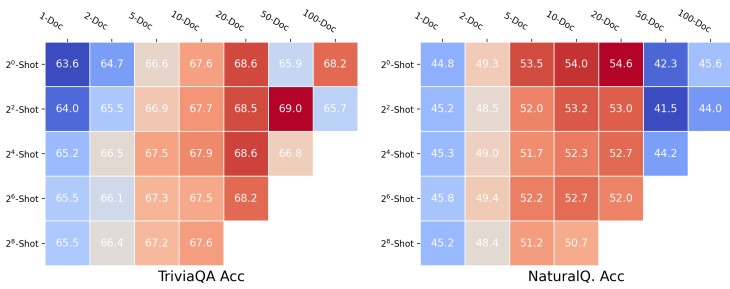

Figure 9: Evaluation accuracy of DRAG on TriviaQA and Natural Questions (NaturalQ.).

In addition to multi-hop question answering datasets, we also report results on one-hop datasets, specifically TriviaQA and Natural Questions (Joshi et al., 2017; Kwiatkowski et al., 2019). The evaluations for one-hop datasets are performed with DRAG and presented in Figure 9, similar to Figure 8. For TriviaQA, increasing the number of documents generally leads to improved accuracy, where the highest accuracy of 69.0% is achieved with 50 documents. In Natural Questions, performance increases with the number of documents up to about 10 or 20 documents, but further increases in the document count lead to diminishing returns or even slight declines in accuracy. The highest accuracy of 54.6% is achieved with 20 documents in 1-shot, and performance drops slightly when

more documents are included. In summary, the optimal number of shots falls between 1 and 4. While increasing the number of shots and documents leads to initial performance gains, these improvements plateau beyond certain thresholds. This trend, in contrast to multi-hop datasets, may be partially attributed to the nature of the one-hop questions and retrieval relevance.

Table 7: StrategyQA accuracy results.

|  | StrategyQA | | | | |
| --- | --- | --- | --- | --- | --- |
|  | Zero-shot QA | Many-shot QA | RAG | DRAG | IterDRAG |
| Acc | 61.1 | 74.7 | 74.7 | 79.0 | 83.4 |

We also include the multi-hop and binary StrategyQA dataset in our experiments, see Table 7 (Geva et al., 2021). Despite being binary questions, we observe similar trends to our main experiments. For example, DRAG consistently outperforms the baseline QA and RAG methods, with 29.3% accuracy improvement to for the baseline QA model. Furthermore, the performance is boosted with 83.4 accuracy using the iterative IterDRAG. These results demonstrate that even for binary, multi-hop tasks, iterative approaches provide substantial gains, confirming the effectiveness of both long-context and iterative strategies for inference scaling in RAG.

# E ADDITIONAL RESULTS ON INFERENCE SCALING LAWS WITH GTR RETRIEVER

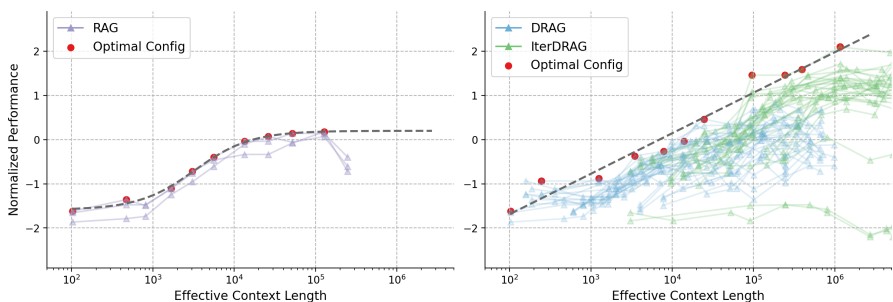

Figure 10: Normalized performance with increasing effective context lengths on MuSiQue, the results are obtained with GTR XXL retriever and Gemini 1.5 Flash.

To enhance the generalizability of our scaling observations and validate the findings with an alternative retriever model, we conduct additional experiments using the open-source GTR XXL retriever and Gemini 1.5 Flash (Ni et al., 2022). Figure 10 shows the results on MuSiQue, evaluated using 100 sampled examples from the dataset for computational efficiency. Different from the performance plateau in standard RAG, DRAG and IterDRAG yields consistent performance gains with increasing context length, especially with IterDRAG at longer context lengths. Overall, the results demonstrate consistent patterns in inference scaling even with a different retriever model, highlighting the potential of expanding text-time compute in long-context RAG for more generalized scenarios.

# F ADDITIONAL RESULTS ON INFERENCE SCALING LAWS FOR RAG

We present data-specific results on the relationship between the performance and the effective context length. Figure 11 presents the results on the other three datasets other than MuSiQue (See Figure 1 for visualized results on MuSiQue). We observe different behavior depending on the datasets. For instance, the gains are more linear and consistent on Bamboogle and MuSiQue, and almost linear on 2WikiMultiHopQA until 1M tokens. However, HotpotQA and 2WikiMultiHopQA with effective context length longer than 100k tokens exhibit more sigmoidal patterns, likely due to the difficulty of the datasets and the quality of the retrieved documents.

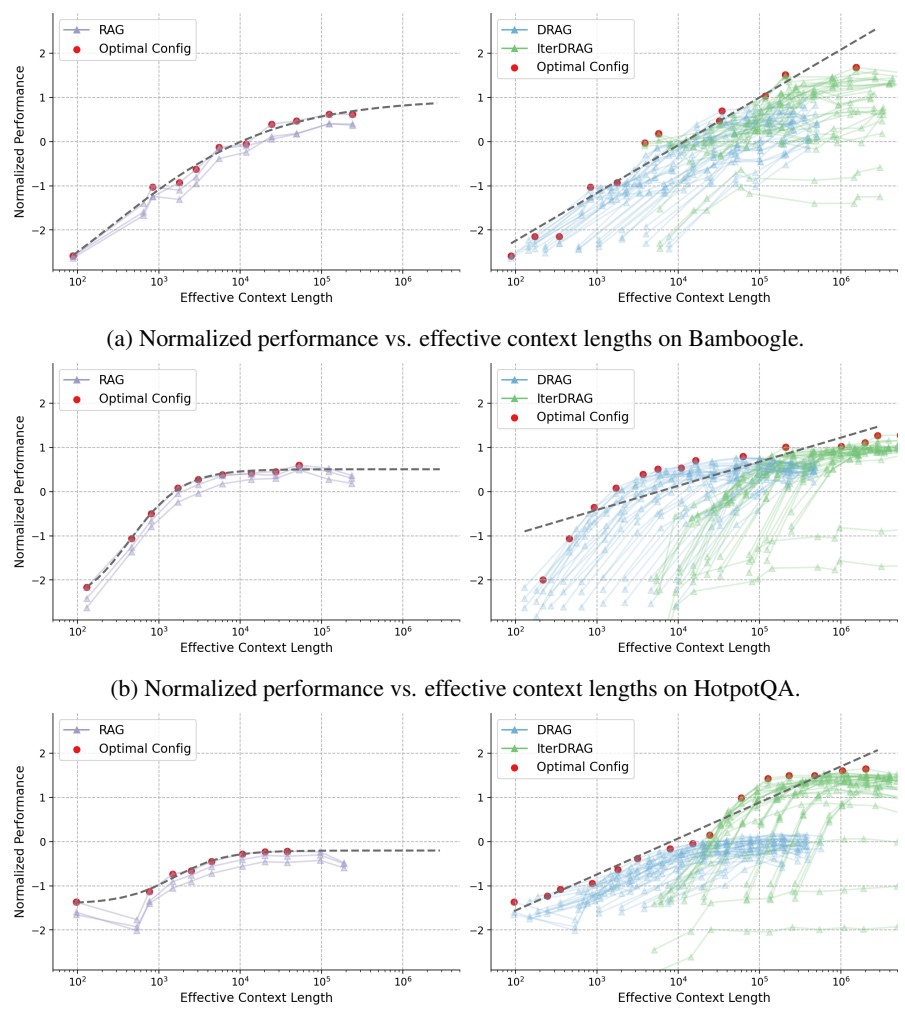

(a) Normalized performance vs. effective context lengths on Bamboogle.

(b) Normalized performance vs. effective context lengths on HotpotQA.

(c) Normalized performance vs. effective context lengths on 2WikiMultiHopQA.

Figure 11: Normalized performance with increasing effective context lengths on different datasets.

## G  ADDITIONAL RESULTS ON COMPUTATION ALLOCATION MODEL FOR RAG

Table 8: Computation allocation mode of Gemini 1.5 Flash with $p$-value, $R^2$ and MSE statistics.

|  | $a$ | | | $b$ | | | $c$ | $R^2$ | MSE |
|---|---|---|---|---|---|---|---|---|---|
| Value | 0.325 | 0.101 | 0.177 | -0.067 | -0.008 | 0 | -0.730 | 0.903 | 0.085 |
| $p$-value | 0.000 | 0.000 | 0.000 | 0.000 | 0.092 | N/A | 0.000 | N/A | N/A |

We further explore the findings on the computation allocation model. In particular, we report the estimated parameters along with $p$-values, $R^2$, and MSE statistics in Table 8. In our implementation, we constrain the last element of $b$, leaving six learnable parameters in total. Our analysis shows that all parameters are statistically significant, except for $b_1$, which has a $p$-value slightly above 0.05. Nonetheless, our experiments suggest that retaining $b_1$ improves generalization in many cases, such as IterDRAG on multi-hop datasets. For sigmoid scaling, we fit a custom function between the predicted $\hat{P}$ and ground truth $P$ values, defined as $\sigma(x) = \frac{3.30}{1+e^{-1.81(x+0.46)}} - 2.18$.

We also visualize the predictions on for IterDRAG across different datasets in Figure 12, where each subplot represents a dataset and each line corresponds to a document setting ($k$). The inference compute is scaled by increasing the number of in-context examples ($m$) and generation iterations ($n$).

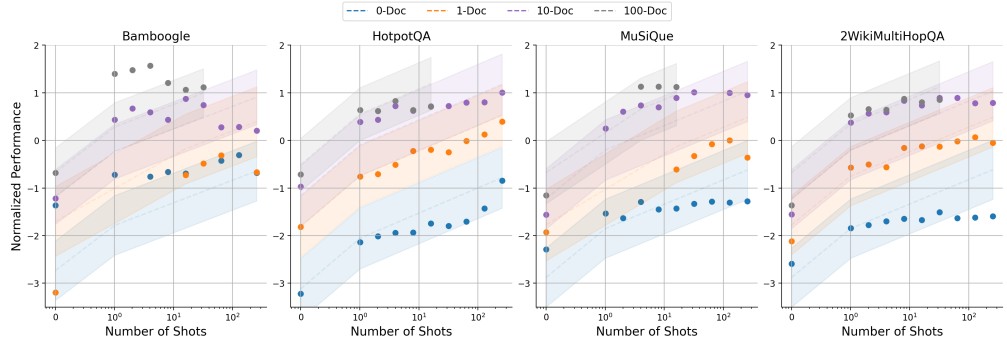

Figure 12: The estimated performance using the proposed computation allocation model vs. actual metric values in IterDRAG. The subplots represent different datasets, where each line corresponds to a fixed number of documents, we scale the context length by increasing the number of shots.

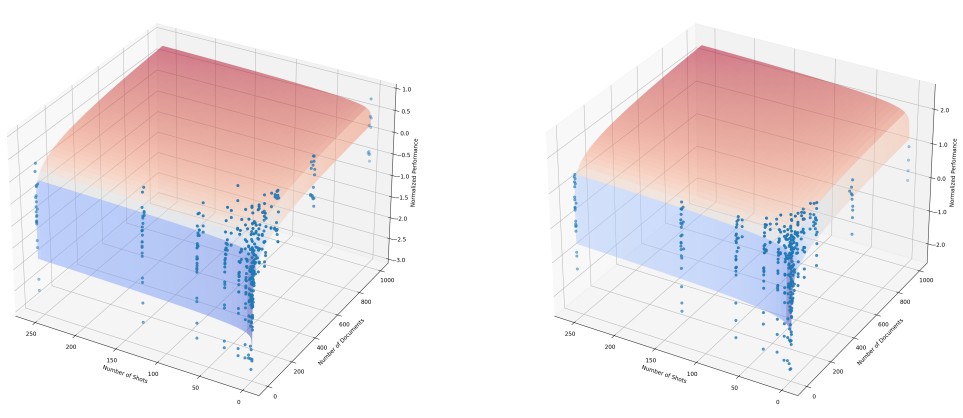

(a) Performance vs. predicted surface for DRAG.    (b) Performance vs. predicted surface for IterDRAG.

Figure 13: Normalized performance vs. predicted surface for DRAG and IterDRAG.

Here, we find similar trends to those in Figure 6, although IterDRAG shows larger variations compared to DRAG. HotpotQA and 2WikiMultiHopQA show more consistent trends with the predictions, likely due to the predominance of multi-hop queries. In summary, our findings are consistent for both DRAG and IterDRAG, demonstrating that RAG performance can be accurately modeled by our computation allocation model for RAG. For Bamboogle, HotpotQA and 2WikiMultiHopQA, we provide the normalized performance with increasing effective context lengths in Figure 11, in which we observe similar trends to the results on MuSiQue (See Figure 1). We also illustrate the prediction surface for both DRAG and IterDRAG in Figure 13.

## H  ERROR ANALYSIS

Despite the performance gains from scaling effective context length, RAG performance on challenging datasets like MuSiQue remain moderate, even for IterDRAG. To address this, we analyze the mistakes in both DRAG and IterDRAG to examine the limitations and errors inherent in these approaches. In the following, we explore common failure cases (See Figure 15) to understand where each method falls short and how they could be further improved.

We provide selected example mistakes from Figure 15a to Figure 15d, with retrieved documents omitted for brevity. The reasons for common errors can be grouped into four categories: (1) inaccurate or outdated retrieval; (2) incorrect or lack of reasoning; (3) hallucination or unfaithful reasoning; and (4) evaluation issues or refusal to answer. We elaborate on these categories below:

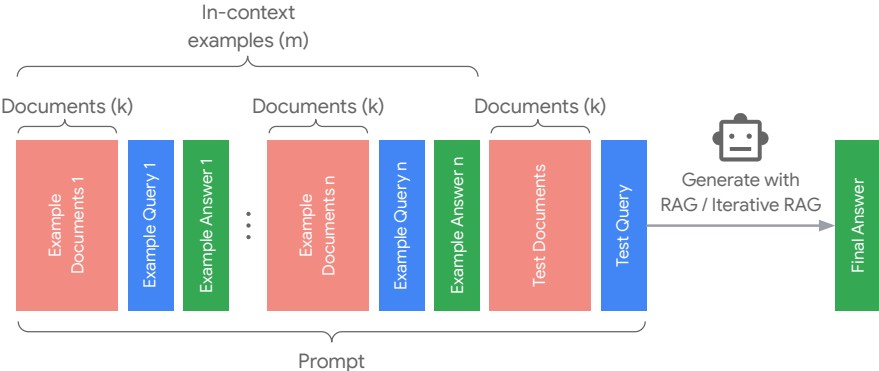

Figure 14: Input prompt that comprises of $m$ in-context examples, the test documents and query, in which each document chunk consists of $k$ retrieved documents. For IterDRAG, the example answers additionally provide sub-queries and intermediate answers as demonstrations.

- **Inaccurate or outdated retrieval**: A major source of RAG errors stems from the retrieval process, where relevant knowledge is not correctly retrieved. For example, in the first question of Figure 15c, the top-50 retrieved documents do not contain the correct answer. A similar issue occurs in the second QA pair, where outdated retrieval results fail to provide useful information. In the third case, although both battles are retrieved, the initial documents overly focus on the Battle of Manila, leading to an incorrect response.

- **Incorrect or lack of reasoning**: Beyond retrieval issues, incorrect reasoning chains are another common source of errors. For example, in the first case in Figure 15b, although the correct documents are retrieved, the reasoning process is incomplete (i.e., no explicit comparison of the mountain heights), leading to an incorrect answer in DRAG. Similarly, in the second and third cases, the reasoning is either absent (as in DRAG) or flawed. As a result, reasoning-related errors tend to occur more frequently in difficult questions and in the one-step DRAG approach.

- **Hallucination or unfaithful reasoning**: Other than retrieval and reasoning, hallucination and unfaithful reasoning also contribute to errors in knowledge-intensive tasks. In the first case, the prediction is incorrect and cannot be found in the retrieved documents. As for the rest cases, while the answers are related, certain steps in the reasoning chain are flawed and cause errors in the final answers. These highlight the persistent challenge of hallucination in LLMs, particularly in long-context generation tasks.

- **Evaluation issues or refusal to answer**: Finally, we observed several evaluation issues that may lead to inaccurate evaluation. For instance, the use of abbreviations or variations in date format can result in incorrect scoring across all metrics. Moreover, our experiments do not account for abstaining from answering, which could cause unfair scores.

## I  IMPLEMENTATION

In our experiments, we utilize the Gecko-1B (en) embedding model to index both the documents and input queries (Lee et al., 2024b), using Wikipedia passages from the KILT benchmark as the document source (Petroni et al., 2020). In test-time, the input query is compared against all embeddings in the corpus, and the top-$k$ neighbors are selected for inference. Each document is then truncated on the right side to a maximum of 1024 tokens using whitespace tokenization. For each example, we arrange the elements in the following order: documents, query, and label, with the retrieved documents listed in reverse order, placing the higher-ranked documents closer to the query (Liu et al., 2024b). Consequently, the prompt comprises of multiple in-context examples, followed by the test documents and test query, as illustrated in Figure 16.

For generation, we utilize Gemini 1.5 Flash for more efficient experiments. In DRAG, inference scaling is achieved by increasing the context length through the combination of documents ($k$) and

in-context examples ($m$). Then, the prompt (See Figure 16a) is provided to the model for a one-time generation using the default generation parameters. For IterDRAG, the input prompt is constructed in a similar fashion, with the example answers consisting of assembled sub-queries, intermediate answers, and the final answer (See Figure 16b). Here, we scale test-time compute by incorporating iterative retrieval and generation, along with the increase of documents and demonstrations. In each iteration, we restrict the generation to adhere to the Self-Ask format, in which the response should start with "Follow up: ", "Intermediate answer: " or "So the final answer is: " (Koo et al., 2024). Each iteration begins with the generation of a sub-query and concludes with the production of an intermediate answer. If a sub-query is generated, additional documents are retrieved and appended to the initial set (i.e., Test Documents in Figure 16), after which the model generates an intermediate answer. We allow up to five iterations, after which the model is forced to produce the final answer.

To evaluate the estimated parameters within computation allocation model for RAG, we normalized the performance metrics by subtracting the mean and dividing by the standard deviation for each dataset and metric. For DRAG, the effective context length is calculated by counting the tokens in the prompt, while for IterDRAG, it is determined by summing the context tokens across all inference requests. We constrain the last parameter in $b$ and perform ordinary least squares to estimate rest six parameters in Equation (2). To prevent numerical instability, we shift the values in $\theta$ by a small constant $\epsilon$ of 0.01. When computing $R^2$ and MSE, we manage noisy data by excluding peak and valley outliers in our experiments. However, for domain generalization and length extrapolation, all data points are included in the evaluation. To predict downstream task performance, $i$ should be computed for each task. Specifically, in each strategy and task: $i_{\text{doc}} = P(k = 1, m = 0, n = 1) - P(k = 0, m = 0, n = 1)$, $i_{\text{shot}} = P(k = 0, m = 1, n = 1) - P(k = 0, m = 0, n = 1)$. For the predicted optimal hyperparameters, we present the actual metric values to validate the efficacy of computation allocation model for RAG.

---

**Inaccurate or outdated retrieval**

`Question:` What is the lowest elevation of the longest railway tunnel?
`Prediction:` 500 meters
`Annotation:` 312 m

`Question:` According to QS World University Rankings, where does the college that Ibrahim Shihata attended rank?
`Prediction:` 3rd
`Annotation:` 551-600

`Question:` Which battle occurred first, the Battle of Manila or the Battle of Guam?
`Prediction:` Battle of Manila
`Annotation:` Battle of Guam

(a) Example mistakes due to inaccurate or outdated retrieval.

---

**Incorrect or lack of reasoning**

`Question:` Which mountain, Masherbrum or Khunyang Chhish, is a taller mountain?
`Prediction:` Masherbrum
`Annotation:` Khunyang Chhish

`Question:` What is the date of death of the director of film The Organization (Film)?
`Prediction:` April 15, 2018
`Annotation:` December 12, 2012

`Question:` Who introduced a system of musical notation in the 14th century that is used in the area where most of the invasion of the eastern Roman Empire took place?
`Prediction:` Philippe de Vitry
`Annotation:` John Kukuzelis

(b) Example mistakes due to incorrect or lack of reasoning.

---

**Hallucination or unfaithful reasoning**

`Question:` Who was the last emperor of the dynasty that succeeded the Song dynasty?
`Prediction:` Emperor Yuanzhen
`Annotation:` Toghon Temür

`Question:` What is another notable work by the illustrator of Sylvester and the Magic Pebble?
`Prediction:` Shrek!
`Annotation:` Doctor De Soto

`Question:` In what movie did a Kenyan-Mexican actress, who graduated from Hampshire College, star in in 2015?
`Prediction:` Queen of Katwe
`Annotation:` Star Wars: The Force Awakens

(c) Example mistakes due to hallucination or unfaithful reasoning.

---

**Evaluation issues or refusal to answer**

`Question:` The most populous city in Punjab is how large (area wise)?
`Prediction:` 310 sq. km
`Annotation:` 310 square kilometers

`Question:` Renáta Tomanová and Larisa Neiland are former professional athletes for what sport?
`Prediction:` Tennis
`Annotation:` Professional tennis

(d) Example mistakes due to evaluation issues or refusal to answer.

Figure 15: Example mistakes of DRAG and IterDRAG across datasets.

**Prompt for DRAG**

You are an expert in question answering. I am going to give you one or more example triples of context, question and answer, in which the context may or may not be relevant to the question. The examples will be written.

Context (which may or may not be relevant):
`<Retrieved documents>`
Question: What is the place of birth of the director of film Servant'S Entrance?
Answer: Helsingfors

`<Further demonstrations>`

After the examples, I am going to provide another pair of context and question, in which the context may or may not be relevant to the question. I want you to answer the question. Give only the answer, and no extra commentary, formatting, or chattiness. Answer the question.

Context (which may or may not be relevant):
`<Retrieved documents>`
Question: Who was born first out of Thomas Henry Holland and Jean-Mandé Sigogne?
Answer:

(a) Example prompt for DRAG. The prompt comprises of instructions and varying number of demonstrations, followed by a test example.

**Prompt for IterDRAG**

You are an expert in question answering. I am going to give you one or more example sets of context, question, potential follow up questions and their respective answers, in which the context may or may not be relevant to the questions. The examples will be written.

Context:
`<Retrieved documents>`
Question: What nationality is the director of film Boggy Creek Ii: And The Legend Continues?
Follow up: Who is the director of the film Boggy Creek II: And The Legend Continues?
Intermediate answer: The director of the film Boggy Creek II: And The Legend Continues is Charles B. Pierce.
Follow up: What is the nationality of Charles B. Pierce?
Intermediate answer: The nationality of Charles B. Pierce is American.
So the final answer is: American

`<Further demonstrations>`

After the examples, I am going to provide another pair of context and question, in which the context may or may not be relevant to the question. I want you to answer the question. When needed, generate follow up question(s) using the format 'Follow up: X', where X is the follow up question. Then, answer each follow up question using 'Intermediate answer: X' with X being the answer. Finally, answer to the main question with the format 'So the final answer is: X', where X is the final answer.

Context:
`<Retrieved documents (with interleaving retrieval)>`
Question: Where was the director of film Death Of A Friend born?
Follow up: | Intermediate answer: | So the final answer is:

(b) Example prompt for IterDRAG. The prompt comprises of instructions and varying number of demonstrations, followed by a test example. In each iteration, we control the generation to follow the Self-Ask format with constrained decoding.

Figure 16: Example prompts for DRAG and IterDRAG.

