# OpenReview forum: "Inference Scaling for Long-Context Retrieval Augmented Generation"
_ICLR.cc/2025/Conference — ICLR 2025 Oral_

### Official Review · Reviewer_FTdU · 2024-10-27

**Soundness:** 4
**Presentation:** 4
**Contribution:** 3
**Rating:** 8
**Confidence:** 4

**Summary:**

This paper systematically investigates the performance of RAG systems as inference computation resources scale up, demonstrating an almost linear improvement in RAG performance with optimal test-time compute allocation. Furthermore, the authors derive an empirical scaling law that can predict the optimal inference parameter configuration for RAG systems under various computational budgets.

**Strengths:**

1. This paper studies a significant issue in the LLM community.
2. The authors conducts extensive experiments across various datasets and evaluation metrics, yielding convincing results.
3. This paper is well structured, clearly written, and easy to understand.

**Weaknesses:**

1. Novelty of the proposed methods is somewhat limited. Both DRAG and IterDRAG are conventional approaches in academia and industry, similar to some classic methods like ITER-RETGEN, IRCoT, and Active-RAG.

2. The experiments were only conducted with Gemini-1.5-Flash and Gecko retrieval. Gemini-1.5-Flash is a relatively small LLM, and different conclusions might be drawn on larger, more powerful LLMs. Moreover, if the scaling laws derived could be generalized to other LLMs and retrieval methods, it would add greater value to this work.

3. Some experimental phenomena lack more in-depth discussion. For example, on line 371, the impact of few-shot examples on IterDRAG should be assessable, at least through ablation studies to determine whether it is the in-context query decomposition or the knowledge extraction capability that is more influential.

4. While increasing the context size can improve RAG performance, it also leads to greater inference time and token consumption, especially when using iterative retrieval. The authors did not discuss this trade-off.

**Questions:**

See Weaknesses.

---

> ### Author Response · Authors · 2024-11-27
> **Response to Reviewer FTdU**
>
> We thank you for your valuable feedback on our submission and are excited that you find our work "convincing", "well structured" and "easy to understand"! We want to clarify your concerns and questions in the following.
>
> *Novelty of the proposed methods is somewhat limited. Both DRAG and IterDRAG are conventional approaches...*
>
> - Thank you for your comment. Existing work like ITER-RETGEN, IRCoT and Active-RAG mostly focus on proposing specific inference strategies to improve RAG performance without systematically understanding the dynamic of how RAG performance changes as inference computation increases.
>
> - In contrast, our primary goal is not to introduce new retrieval or generation strategies, but to systematically understand and model inference scaling for long-context RAG. To this end, we build on existing paradigms and explore combinations of different strategies (via in-context examples, iterative retrieval with constrained decoding etc.) to more effectively scale inference computation. Using these strategies, we demonstrate that RAG performance can scale almost linearly with increasing magnitude of inference computations when optimally configured, as opposed to the sigmoidal trend as in previous studies [1-2]. Additionally, a key contribution of our work is to quantify the relationship between RAG performance and different combinations of inference parameters using the computation allocation model, deriving near-optimal configurations across diverse scenarios.
>
> - Consequently, this distinguishes our work from others by: (1) focusing on the understanding of inference scaling in RAG; and (2) identifying practical solutions to leverage such scaling dynamics in long-context RAG performance.
>
> *The experiments were only conducted with Gemini-1.5-Flash and Gecko retrieval...*
>
> - We thank you for your suggestion. We provide additional results with GTR XXL, detailed in Appendix E of our rebuttal revision. Overall, we observe similar trends with GTR-retrieved documents, showing that long-context RAG performance consistently improves with effective context lengths. Yet due to resource limitations, we are unable to perform large-scale experiments with further LLMs. As such, we leave further exploration across a broader range of models as future work.
>
> *Some experimental phenomena lack more in-depth discussion...*
>
> - Thank you for suggesting the need for a more in-depth discussion on the impact of few-shot examples in IterDRAG. As shown by the green line in Figure 5c, IterDRAG can outperform DRAG even with fewer in-context examples (i.e., less demonstrations for in-context knowledge extraction), where we attribute such performance gains to the additional query decomposition process. To provide more concrete results and a comprehensive analysis of IterDRAG configurations, we have updated our observations and discussion in the rebuttal revision. Please refer to the revised Sec 4.4 and the additional ablation studies in Appendix D. These findings further highlight the varying effectiveness of in-context demonstrations and query decomposition (evidenced, for example, through the heatmap analysis).
>
> *While increasing the context size can improve RAG performance, it also...*
>
> - Thank you for pointing out the trade-off between RAG performance and the associated inference time and token consumption. In our experiments, we quantify inference computation by the number of input tokens across LLM calls (i.e., effective context length), and we have explicitly modeled such trade-off relationships between performance and test-time compute using our computation allocation model (e.g., as shown in  Figure 1 and Figure 4).
>
> - One of the main discoveries of this paper is that there exists a better trade-off between RAG performance and the inference computation from existing strategies: (1) Existing works that only use one strategy to scale up inference computation (e.g., solely adding more documents or demonstrations) have a limitation: beyond a certain threshold, increasing the inference computation (e.g., by adding 10x more documents) no longer improves RAG performance, which is not an ideal trade-off to consume more tokens. (2) Our work demonstrates that by simply employing a combination of multiple inference strategies, further increasing the inference computation can continue to improve RAG performance almost linearly when the optimal inference parameters are identified, leading to a better trade-off than existing work.
>
> [1] Xu, Peng, et al. "Retrieval meets long context large language models." arXiv preprint arXiv:2310.03025 (2023).
>
> [2] Leng, Quinn, et al. "Long Context RAG Performance of Large Language Models." arXiv preprint arXiv:2411.03538 (2024).

---

> > ### Comment · Reviewer_FTdU · 2024-12-02
> >
> > Thanks for your response. Most of my concerns are properly addressed, so I have raised my score to 8.

---

> > > ### Author Response · Authors · 2024-12-03
> > > **Response to Reviewer FTdU**
> > >
> > > Thank you again for your insightful and constructive feedback! We’re excited to hear that our revisions have effectively addressed your concerns!

---

### Official Review · Reviewer_Tea3 · 2024-11-02

**Soundness:** 3
**Presentation:** 3
**Contribution:** 3
**Rating:** 8
**Confidence:** 2

**Summary:**

The paper explores inference scaling for retrieval augmented generation (RAG) in long-context large language models (LLMs), focusing on strategies beyond simply expanding the knowledge base. The authors investigate how in-context learning and iterative prompting can optimize the use of additional compute resources during inference. They address two primary questions: the benefit of inference computation scaling for RAG and the prediction of optimal compute allocation within a given budget. Their findings reveal that optimal allocation of inference computation results in nearly linear performance gains for RAG, a phenomenon described as inference scaling laws. The authors also develop a computation allocation model that accurately predicts the optimal inference parameters, with experimental results showing up to 58.9% performance improvements on benchmark datasets compared to standard RAG configurations.

**Strengths:**

The research question is quite interesting, as there is not much work on inference time scaling for RAG; this study systematically explores this area and may draw some attention.

**Weaknesses:**

I am concerned that this work is more suitable as a technical report rather than a research-oriented study. There is considerable related work combining long-context LLMs and RAG, and the main contribution of this work is mainly the proposed RAG inference scaling law. However, this conclusion is method-specific and may not apply to other methods.

**Questions:**

1. Can you provide a straightforward explanation of your findings and how they guide the use of long-context LLMs for RAG?
2. If other prompts or methods are used, is your computation allocation model still applicable?

---

> ### Author Response · Authors · 2024-11-27
> **Response to Reviewer Tea3**
>
> We appreciate your insightful comments on our work and are thrilled that you find our submission "interesting" and "systematic"! We want to clarify your concerns and questions below.
>
> *I am concerned that this work is more suitable as a technical report rather than a research-oriented study...*
>
> - We appreciate the concern regarding the scope of our work, yet we believe it extends beyond a technical report by providing a structured, empirical framework for understanding inference scaling in long-context RAG [1-2]. Unlike previous works [3-4] that focuses on one specific RAG strategy (e.g., increasing the number of documents or the length of documents), our study provides a comprehensive understanding of inference scaling for RAG by exploring various inference strategies across different compute budgets (measured by effective context lengths). Our experiments reveal a new scaling trend / dynamic, i.e., that RAG performance can scale almost linearly with increased magnitude of inference computation instead of sigmoidal as shown in previous studies [4], as long as one uses the right combination of inference parameters. To identify the "right" inference configuration, we also propose the computation allocation model to quantitatively model the RAG performance across different combinations of inference parameters, offering near-optimal configurations for various scenarios. Considering the above contributions, our work lays a foundation for future research in scaling inference compute in long-context RAG, offering insights that extend beyond a technical report.
>
> *Can you provide a straightforward explanation of your findings and how they guide the use of long-context LLMs for RAG?*
>
> - In summary, our study investigates inference scaling for long-context RAG, demonstrating that performance can scale nearly linearly with increasing inference compute. Building on these findings, we propose the computation allocation model, which derives near-optimal inference parameters for various knowledge-intensive tasks.
>
> - More specifically, we conduct an extensive evaluation of long-context RAG performance across diverse inference configurations and present the following findings: (1) Unlike previous studies showing that RAG performances stop increases  when scaling the amount of retrieval information beyond a certain threshold, our findings reveal that RAG performance can continue to increase and scale almost linearly with increased inference computation given optimal inference parameters (e.g., in-context examples and multi-step reasoning). (2) By introducing the computation allocation model, we then provide a systematic framework to predict the right configuration of inference strategies for given budgets, enabling efficient and effective use of long-context LLMs for knowledge-intensive tasks.
>
> *If other prompts or methods are used, is your computation allocation model still applicable?*
>
> - For improved generalizability, we include supplementary experiment results using the GTR XXL retriever model, as presented in Appendix E in the rebuttal revision. In sum, similar observations are made with GTR-retrieved documents, showing that long-context RAG performance consistently improves with effective context lengths.
>
> - While there are numerous ways to scale inference compute for RAG (e.g., more documents), we focus on commonly used methods such as expanding documents, many-shot demonstrations and iterative prompting, refining these approaches for effective long-context scaling. Nonetheless, our main focus is to show that inference scaling and the modeling of such scaling is feasible for long-context RAG, yielding consistent gains with increased computation budgets. Our approach can be easily adapted for further prompt designs / model families, providing a framework to evaluate and understand the inference scaling properties of both existing and emerging RAG strategies.
>
> [1] Wu, Yangzhen, et al. "An empirical analysis of compute-optimal inference for problem-solving with language models." (2024).
>
> [2] Ruan, Yangjun, Chris J. Maddison, and Tatsunori Hashimoto. "Observational Scaling Laws and the Predictability of Language Model Performance." arXiv preprint arXiv:2405.10938 (2024).
>
> [3] Xu, Peng, et al. "Retrieval meets long context large language models." arXiv preprint arXiv:2310.03025 (2023).
>
> [4] Jiang, Ziyan, Xueguang Ma, and Wenhu Chen. "Longrag: Enhancing retrieval-augmented generation with long-context llms." arXiv preprint arXiv:2406.15319 (2024).

---

> > ### Author Response · Authors · 2024-12-03
> > **Reminder for Discussion**
> >
> > Thank you once again for your thoughtful feedback, we truly appreciate the time and effort you have dedicated to reviewing our work. As the discussion phase is ending, please don’t hesitate to let us know if you have any additional questions or concerns regarding our work!

---

### Official Review · Reviewer_Xv2A · 2024-11-03

**Soundness:** 3
**Presentation:** 3
**Contribution:** 3
**Rating:** 8
**Confidence:** 3

**Summary:**

This paper explores inference scaling in long-context RAG; it analyses downstream QA results in different configurations, showing that RAG performance improves ~linearly with increasing test-time compute under optimal inference parameters. Furthermore, based on these observations, authors derive a set of inference scaling laws for RAG and a computation allocation model.

My only concern is to what extent the current model generalises -- from Fig. 4b, it seems like the considered models might suffer from "lost in the middle" (https://arxiv.org/abs/2307.03172) problems, while more recent long-context models seem to suffer from this significantly less (e.g., https://arxiv.org/abs/2404.16811)

**Strengths:**

Inference-time scaling laws for RAG systems -- extremely interesting, and the community really needs an analysis like this one.

**Weaknesses:**

It is not clear whether the current analysis may generalise to future SFT/RLHF regimens.

**Questions:**

More recent models may suffer less from "lost in the middle" issues -- does the current analysis still hold?

---

> ### Author Response · Authors · 2024-11-27
> **Response to Reviewer Xv2A**
>
> We thank you for your valuable feedback on our work, we are particularly excited that our contributions are regarded as "extremely interesting"! We hope to address your concerns and questions in the following.
>
> *It is not clear whether the current analysis may generalize to future SFT/RLHF regimens.*
>
> - While our computation allocation model and scaling strategies are designed primarily for long-context retrieval augmented generation during inference, similar settings could be applied to SFT/RLHF settings for improved long-context RAG performance. For example, for a fixed training token budget, one can conduct experiments to figure out the optimal allocation of different tasks in the training data. While this falls outside the scope of this paper which focuses more on inference scaling, we agree that this would be a promising future direction to look into.
>
> *More recent models may suffer less from "lost in the middle" issues -- does the current analysis still hold?*
>
> - Thank you for pointing out this. Gemini 1.5 (as used in our experiments) is one of the recent models that demonstrates improved performance of "lost in the middle" / long-context modeling [1, 2]. However, we still notice that RAG performance plateaus when merely increasing the number of retrieved documents. Experiments from [3] also show similar observations using a few other recent models with alleviated "lost in the middle" issue. Therefore, solely addressing "lost in the middle" (i.e., enhancing model’s ability to find relevant information in the context) may not result in a linear scaling of RAG performance with respect to the magnitude of inference computation. It is also crucial to enhance the model’s ability to integrate and reason over the "found relevant information". Consequently, we combine multiple additional scaling strategies such as adding demonstrations and iterative querying and show that it is possible to achieve near-linear performance gains in long-context RAG as the magnitude of effective context length increases.
>
> [1] Reid, Machel, et al. "Gemini 1.5: Unlocking multimodal understanding across millions of tokens of context." arXiv preprint arXiv:2403.05530 (2024).
>
> [2] An, Shengnan, et al. "Make Your LLM Fully Utilize the Context." arXiv preprint arXiv:2404.16811 (2024).
>
> [3] Leng, Quinn, et al. "Long Context RAG Performance of Large Language Models." arXiv preprint arXiv:2411.03538 (2024).

---

> > ### Comment · Reviewer_Xv2A · 2024-11-29
> >
> > Thanks!

---

> > > ### Author Response · Authors · 2024-12-03
> > > **Response to Reviewer Xv2A**
> > >
> > > Thank you again for your valuable and thoughtful feedback. Please don’t hesitate to reach out if you have any further questions or concerns!

---

### Official Review · Reviewer_RvZP · 2024-11-11

**Soundness:** 3
**Presentation:** 4
**Contribution:** 3
**Rating:** 8
**Confidence:** 3

**Summary:**

This paper studies the inference scaling behaviors of two retrieval augmented generation (RAG) methods, demonstration-based RAG (DRAG) and iterative demonstration-based RAG (IterDRAG). The inference computation can be scaled in multiple ways, including increasing the number of retrieved documents, in-context examples, or introducing additional generation steps in IterDRAG. Experimental results show DRAG and IterDRAG achieve scaling properties with the proposed configurations, and demonstrate the performance of DRAG and IterDRAG can scale almost linearly with an increasing computation budget. Besides, the paper also learns a computational allocation model that could provides configuration guidance for DRAG and IterDRAG.

**Strengths:**

The paper studies two interesting research questions including the scaling behavior and the prediction of test-time computation allocation long-context RAG methods. The paper conducts systematical experiments on inference scaling of long-context RAG models, and reveals the scaling properties of DRAG and IterDRAG, i.e., the performance improves almost linearly with optimal configuration. Besides, the computational allocation model generalizes well across domains and context lengths, which potentially helps the community to better configure RAGs.

**Weaknesses:**

I have a question on the application of the computational allocation model. When pretraining LLMs, computational allocation models are crucial since pretraining is extremely resource-intensive. However, inference is typically much less costly by comparison. So, why not determine the best configuration by simply searching it?

**Questions:**

Please see the question above.

---

> ### Author Response · Authors · 2024-11-27
> **Response to Reviewer RvZP**
>
> We appreciate your thoughtful feedback and are glad that our contributions are considered "interesting" and "systematic". Regarding your question on our computation allocation model, we agree that grid search is feasible for smaller datasets when context length is limited. However, as context length increases or with larger evaluation sets, grid search becomes extremely expensive. For instance, we estimate that conducting a grid search on MuSiQue for an effective context length budget of 5M tokens using Gemini 1.5 Flash could cost up to $18,879, even with sub-sampling from the evaluation set. While inference generally demands less computation than pretraining, directly searching for the best configuration without guidance can still lead to inefficiencies due to the vast space of inference hyperparameters. Our computation allocation model addresses this challenge by systematically modeling inference compute, delivering near-optimal configurations based on the estimated relationship between performance and inference parameters. This approach not only optimizes long-context RAG solutions within a given compute budget but also generalizes well across tasks and constraints, eliminating the need for costly and exhaustive grid searches during evaluation.

---

> > ### Author Response · Authors · 2024-12-03
> > **Reminder for Discussion**
> >
> > Thank you once again for your insightful and constructive feedback. We are grateful for the time and effort you have dedicated to reviewing our work. As the discussion phase comes to an end, kindly let us know if you have any further questions or concerns:)

---

### Author Response · Authors · 2024-11-27
**Response to All Reviewers**

We sincerely thank the chairs and reviewers for their reviewing efforts and the constructive suggestions on our work, which are invaluable in refining our paper. In response to reviewers' comments, we have thoroughly revised our manuscript and incorporated additional analyses and experiments. Our rebuttal revision includes the following updates:

- Improved the writing and clarity of the introduction, adopted methods, inference scaling, and analysis in the initial sections.

- Updated the analysis of inference scaling and parameter-specific scaling (e.g., examining the varying effectiveness of different inference parameters, see Sec. 4.3 and Sec. 4.4).

- Added discussion on the trade-off between RAG performance and inference budget, and relocated the discussion section to Appendix A.

- Included additional experimental results using GTR XXL in Appendix E.

- Restructured the appendix into sections for improved organization and readability.

In addition to the updated manuscript, we have provided detailed responses to reviewers' concerns and questions directly alongside each review. We remain available and are happy to address any further questions or feedback you may have. Once again, we deeply appreciate the time and effort you dedicated to reviewing our work, and we are grateful for the opportunity to improve our research with your insights!

Best

Authors

---

### Meta-Review · Area_Chair_Bbsf · 2024-12-20

**Metareview:**

This paper presents a detailed investigation into the inference scaling of retrieval augmented generation (RAG) for long-context LLMs, exploring effective strategies for utilizing external knowledge beyond merely increasing its quantity. The authors focus on in-context learning and iterative prompting as scalable strategies to optimize test-time computation, addressing how these approaches enhance LLM performance and developing a model to predict optimal computation allocations. The findings indicate nearly linear gains in RAG performance with optimal computation scaling, substantiated by a novel computation allocation model that accurately predicts the best settings under various constraints.

The reviewers are unanimous in their strong support for this work, commending its insightful analysis, substantial performance improvements on benchmarks, and its potential to significantly advance the field of long-context LLMs, leading to a recommendation for acceptance.

**Additional Comments On Reviewer Discussion:**

Nil

---

### Decision · Program_Chairs · 2025-01-22

Accept (Oral)